# Cross-Talk between Neurohormonal Pathways and the Immune System in Heart Failure: A Review of the Literature

**DOI:** 10.3390/ijms20071698

**Published:** 2019-04-05

**Authors:** Elena De Angelis, Michela Pecoraro, Maria Rosaria Rusciano, Michele Ciccarelli, Ada Popolo

**Affiliations:** 1Department of Medicine, Surgery and Odontology, University of Salerno, via S.Allende 1, 84081 Baronissi (SA), Italy; elena.deangelis2@gmail.com (E.D.A.); myra80@gmail.com (M.R.R.); mciccarelli@unisa.it (M.C.); 2Department of Pharmacy, University of Salerno, via Giovanni Paolo II 132, 84084 Fisciano (SA), Italy; mipecoraro@unisa.it; 3Casa di Cura Montevergine, 83013 Mercogliano (AV), Italy

**Keywords:** heart failure, immune system, adrenergic system, renin-angiotensin-aldosterone system, natriuretic peptides system, neuro-immunomodulation, heart failure with preserved ejection fraction

## Abstract

Heart failure is a complex clinical syndrome involving a multitude of neurohormonal pathways including the renin-angiotensin-aldosterone system, sympathetic nervous system, and natriuretic peptides system. It is now emerging that neurohumoral mechanisms activated during heart failure, with both preserved and reduced ejection fraction, modulate cells of the immune system. Indeed, these cells express angiotensin I receptors, adrenoceptors, and natriuretic peptides receptors. Ang II modulates macrophage polarization, promoting M2 macrophages phenotype, and this stimulation can influence lymphocytes Th1/Th2 balance. β-AR activation in monocytes is responsible for inhibition of free oxygen radicals production, and together with α2-AR can modulate TNF-α receptor expression and TNF-α release. In dendritic cells, activation of β2-AR inhibits IL-12 production, resulting in the inhibition of Th1 and promotion of Th2 differentiation. ANP induces the activation of secretion of superoxide anion in polymorphonucleated cells; reduces TNF-α and nitric oxide secretion in macrophages; and attenuates the exacerbated TH1 responses. BNP in macrophages can stimulate ROS production, up-regulates IL-10, and inhibits IL-12 and TNF-α release by dendritic cells, suggesting an anti-inflammatory cytokines profile induction. Therefore, different neurohormonal-immune cross-talks can determine the phenotype of cardiac remodeling, promoting either favorable or maladaptive responses. This review aims to summarize the available knowledge on neurohormonal modulation of immune responses, providing supportive rational background for further research.

## 1. Introduction

Heart failure (HF) is a complex clinical syndrome that occurs when the heart is unable to pump blood at a rate adequate to the requirements of the metabolizing tissues [1]. In fact, HF can be differentiated according to the ejection fraction in HF with preserved ejection fraction (HFpEF; EF ≥ 50%), with reduced ejection fraction (HFrEF; EF < 40%) and with mid-range ejection fraction (HFmrEF; EF = 40%–49%) [2]. HFpEF has been identified only recently as clinical pathology, despite it account for ~40% of all heart failure cases, and being responsible for the majority of hospital admissions related to heart failure with in-hospital mortality only slightly lower than that in patients with HFrEF [1]. Preserved EF does not imply that systole is normal, and indeed it has been recently observed using tissue Doppler speckle tracking that HFpEF patients had reduced longitudinal and circumferential strain [3,4].

HFrEF and HFpEF, present different etiologies, pathogenetic mechanisms, and treatments.

Several pathogenetic mechanisms are involved in HFrEF pathogenesis: dysregulated neurohumoral stimulation, increased hemodynamic overload, ischemia-related damage, ventricular remodeling, immunological stimulation, abnormal myocyte calcium cycling, extracellular matrix anomalies, accelerated apoptosis, and genetic mutations [1]. 

In HFpEF, the diastolic dysfunction plays a pivotal role, and its pathogenesis is prevalently caused by the accumulation of extracellular matrix, as a consequence to the endothelial to mesenchymal transition (EndMT) and activation of the renin-angiotensin-aldosterone system [5,6]. In patients with HFpEF, the increase in macrophages within the myocardium and during systemic pro-inflammation is likely mediated by mineralocorticoid receptor activation and other pro-inflammatory conditions, including obesity and diabetes, which have been previously demonstrated [5,7].

A multitude of neurohormonal pathways, which serve as the first mechanism of compensation, are undoubtedly involved in the pathogenesis of HFrEF: renin angiotensin aldosterone system (RAAS), the sympathetic nervous system (SNS), and the natriuretic peptides system. However, the chronic activation of RAAS and SNS in particular contributes to disease progression by promoting cardiac remodeling and worsening of function.

The excessive sympathetic stimulation leads to desensitization and down-regulation of the β adrenergic receptors (ARs), resulting in decreased left ventricle contractility and the development of a dilated cardiomyopathy phenotype. Furthermore, norepinephrine release stimulates the activation of the renin-angiotensin-aldosterone nervous system through the release of renin from specialized cells within the juxtamedullary apparatus of the kidneys [8]. The upregulation of RAAS—inducing sodium and water retention, as well as vasoconstriction—is counter-regulated by the natriuretic peptide system, which is stimulated by the elevation of wall stress contributing to the inhibition of arginine-vasopressin secretion, modulating the autonomic nervous system, promoting natriuresis and vasodilation [9]. These neuro-hormonal pathways play an important role also in HFpEF pathogenesis, although it is less known than in HFrEF.

Beyond these canonical mechanisms, neurohormonal pathways interplay through the innate immune system to address the phenotype of cardiac remodeling. For example, after an acute myocardial infarction, activation of the innate immune system is a prerequisite for adequate healing, while a long-term chronic innate immune activation becomes detrimental resulting in adverse left ventricular remodeling and aggravation of heart failure. Cardiac remodeling involves cardiomyocytes, with the induction of pathways leading to hypertrophy, and also the interstitial cells, fibroblasts, collagen, and coronary vasculature that are all involved in myocardial fibrosis. Increased neurohumoral activation, together with cytokine activation, stimulates collagen synthesis leading to fibrosis and remodeling of the extracellular matrix [10]. TGF-β and AngII trigger EndMT-generating cells that still express endothelial markers while gaining fibroblast-like characteristics [11]. This process, well-studied in cardiovascular development, through TGF-β, BMP, and Notch signaling, is involved in the development of cardiac valves and septa in embryonic stages [12,13,14,15]. EndMT is also involved in neovascularization and scar formation in the infarcted tissues where it preserves heart function and slows down the occurrence of heart failure [16,17]. Furthermore, it has been shown that EndMT is associated with the development, progression, and plaque stability of atherosclerosis contributing to neointimal hyperplasia and promoting atherogenic differentiation of endothelial cells under disturbed fluid shear stress [18].

Recent evidences display that angiotensin II receptor antagonist, losartan, suppresses cardiac fibrosis by blocking EndMT via TGF-β/Smad2 [19,20].

The phenomena of cardiac remodeling, in addition to several signaling and cellular cross-talks regulate the process of cardiac fibrosis. In this review, we focused on the role of neurohormonal systems in the pathophysiology of heart failure, both with preserved EF and with reduced EF, through the regulation of immune cells.

## 2. Mechanisms of Immune-Modulation in HFrEF

The cross-talk between the immune system and neurohormonal activation has been deeply investigated. Indeed, RAAS, SNS, and the natriuretic peptide system interfere with inflammatory processes by modulating both the innate and the adaptive immune system.

### 2.1. Renin-Angiotensin Aldosterone System (RAAS)

The RAAS system is a critical hormonal signaling cascade engaged in body fluid and blood pressure (BP) homeostasis [21]. RAS influences BP by regulating salt and water balance, and vasoconstriction, but inappropriate activation of this system leads to cell/tissue remodeling and dysfunction of the cardiovascular system.

The major primary effector of RAAS is the hormone Angiotensin II (Ang II), a potent vasopressor octapeptide formed from sequential proteolysis of its precursor, angiotensinogen (AGT), within the extracellular space. In the first rate-limiting step in Ang II formation, AGT circulating in plasma is proteolyzed into the inactive decapeptide, angiotensin I (Ang I), by an aminopeptidase (Renin) secreted from specialized juxtaglomerular cells of the kidney [22]. Angiotensin-I is rapidly converted to Ang II by the angiotensin-converting enzyme (ACE)—this occurs primarily in the lungs [23].

The "alternative RAAS axis" produces Ang1–7 via subsequent cleavage of AngII by the ACE homolog ACE2. All Angiotensin peptides bind to G protein-coupled receptors (GPCRs), namely AT1 and AT2 receptors for AngII, and Mas receptor for Ang (1–7), with the ability to activate distinct signaling pathways leading to different and often opposite cellular effects [24]. Indeed, the effects of AngII on cardiovascular, renal, and cerebral functions are mediated through the activation of angiotensin type 1 (AT1) receptors [25], but these actions are counteracted by activation of the AT2 receptor and "protective" ACE2/Ang(1-7)/Mas axis [24]. Although AngII is the classic effector molecule of RAS, several RAS enzymes affect immune homeostasis independently of canonic angiotensin II generation and contribute to AngII-mediated end-organ damage.

AT1 receptors expressed in the cardiovascular system activate a variety of intracellular protein kinases including mitogen-activated protein kinase (MAPK) family, p70 S6 kinase, AKT/protein kinase B (PKB), various protein kinase C (PKC) isoforms, receptors, and non-receptor tyrosine kinases and serine/threonine kinases [26,27,28,29,30]. These kinases stimulate NADPH oxidase, ROS generation, and protein synthesis, causing hypertrophy, hyperplasia, and migration of vascular smooth muscle cells (VSMCs), cardiac hypertrophy and renal deterioration. Furthermore, AngII activates the Rho/Rho-associated protein kinase (ROCK) pathway involved in vascular remodeling and cardiovascular diseases more than in NF-κB activation and subsequent IL-6 expression in VSMCs [31].

AngII induces the release of Aldosterone by the adrenal gland. Aldosterone can also signal directly within the myocardium via the resident minaralcorticoid receptor, thus inhibiting nitric oxide synthase and promoting inflammation, fibrosis, and cardiac myocyte apoptosis [32].

In addition to the well-established role in blood pressure regulation and direct cardiac damage, Ang II also mediates blood pressure-independent effects leading to target organ damage through various inflammatory processes [33], playing a pivotal role in the process of cardiac remodeling [34].

In the adult organism, type 1 angiotensin receptor (AT1) is expressed in cells of the immune system and in particular on macrophages, and T and B lymphocytes that are primarily involved in the development and progression of HF [34]. In vitro, Ang II stimulates the proliferation of splenic lymphocytes [35]. The AT2 receptor is not highly expressed on cells of the immune system [36], but has been detected on human T and NK cells; Ang II and its precursors, angiotensinogen and Ang I, are capable of inducing human T lymphocyte and Natural Killer cell (NK) proliferation [37].

Ang II triggers vascular damage by inducing the activity of adhesion molecules, recruiting inflammatory cells, modulating cytokine expression, and repairing tissue [34]. Tissue-specific regulation and functions of the RAAS are evident in organs engaged in innate and adaptive immune responses.

Several shreds of evidence have reported a connection between Ang II and two pivotal mediators for heart remodeling, the cytokines transforming growth factor (TGF)-β and the tumor necrosis factor (TNF)-α [38,39].

In cardiac fibroblasts, AngII induces the expression of TGF-β through AT1R, expression of collagen through TGF-β/SMAD pathway, and extracellular signal-regulated kinase via an IL-6-dependent mechanism [40]. TNF-α and IL-1β accentuate the effect of TGF-β by driving epithelial to mesenchymal trans-differentiation.

In vitro studies demonstrate that Ang II infusion stimulates the production of IL-1β, IL-6, and inducible nitric oxide synthase (iNOS) [34].

Interestingly, IL-1β and TNF-α act coordinately to increase AT1 receptor density in post-myocardial infarction, thus suggesting a vicious cycle of Ang II-induced myocardial cytokine production and cytokine-induced increase in Ang II activity [41].

Ang II modulates macrophage polarization suppressing, through the AT1 receptor, the M1-macrophage phenotype via proinflammatory activity, and promoting M2 macrophages which inhibit inflammation, promote cell proliferation, and stimulate angiogenesis, via reductions in TNF and IL-1β expression [34].

Ang II is also responsible for altering the Th1/Th2 balance by increasing the production of Th1 cytokine IFN-γ, showing a pro-inflammatory action, and a decrease in that of the Th2 cytokine IL-4 [42]. AT1 receptor stimulation on T cells suppresses T-bet-dependent differentiation of CD4+ T helper cells toward the pro-inflammatory Th1 cell lineage [21].

Activation of AT1 receptors can directly act on hematopoietic cells providing an immunosuppressive signal to temper or limit the pathogenic actions of inappropriate RAAS activation in other organs and tissues [21].

Collectively, the role of the RAAS system in modulating the immune system can have different implications in the context of cardiac remodeling. These multiple facets may not be considered exclusively detrimental to the cardiovascular system. ATR stimulation can enhance cytokine production, as well as macrophage polarization and migration with increased cardiac fibrosis, but AT1R agonism on hematopoietic cells could represent the key for inhibitory feedback attenuating the adverse effects of the RAAS system during HF (Figure 1).

### 2.2. Nervous Sympathetic System

The SNS regulates cardiovascular responses through the activation of the α- and β-adrenergic receptors (AR) by epinephrine (Epi), released by the adrenal medulla, and norepinephrine (NEpi), also released by the adrenal medulla as well as from sympathetic nerve endings [23]. Nonetheless, the release of NEpi and Epi stimulates directly, the activation of both α- and β-adrenergic receptors expressed on immune cells, but also indirectly they regulate cytokine release and stromal cell function. The SNS is fundamental for maintaining immune system homeostasis, augmenting host defense to eliminate pathogens, promoting healing after tissue injury, and tissue repairing after the pathogen elimination.

Under physiological conditions, these responses are adaptive, but in pathological conditions like heart failure, they become maladaptive [43,44].

#### 2.2.1. Adrenergic Receptors (AR)

The AR family includes nine AR subtypes (three α-1, three α-2) and three β-AR subtypes (β1, β2, β3), all belonging to the class of GPCR [45,46]. They consist of a seven-transmembrane-spanning receptor and are coupled to an intracellular heterotrimeric G-protein complex whose signaling plays a critical role in the regulation of multiple functions and processes including those of the immune cells. Each type of AR activates a specific signaling pathway, resulting in receptor-specific immunomodulatory effects.

The β2-AR is the prevalent subtype expressed by immune cells.

For a long time, it was thought that this receptor led only to cAMP–PKA pathway activation with immunosuppressive effects. Evidence from non-immune organs highlighted novel alternative signaling mechanisms induced by β2-AR activation, such as the switch from cAMP–PKA to mitogen-activated protein kinase (MAPK) pathway. This leads to multiple signal transduction pathways activation, resulting in both inhibitory and/or activating effects on cell functions.

This process is the result of β2-AR phosphorylation by PKA and transiently by G protein-coupled kinases (GRKs). The specific subtype of GRK that phosphorylates the receptor determines the functional role of β-arrestin.

Arrestins are a small family of proteins that regulate signal transduction of G protein-coupled receptors [47]. Mammals express four arrestin subtypes, two of which are ubiquitously expressed in the mammalian cardiovascular system, β-arrestin-1 (is the more abundant) and β-arrestin-2, also known as Arrestin-2 and -3, respectively [48]. The principal role of Arrestin-2 and -3 is to desensitize GPCRs, i.e., uncoupled them from G proteins, and subsequently internalize the receptor via clathrin-coated pits, serving as a signalosome that transduces signals in the cytoplasm [49]. Phosphorylation by GRK2 induces β-arrestin-mediated desensitization, whereas phosphorylation by GRK-5 or -6 results in β-arrestin-mediated signaling [50]. The PKA and GRK-2-mediated phosphorylation of different serine sites on the carboxy tail of the β2-AR has two significant consequences: the switch of receptor coupling from Gs, "turned off," to Gi proteins, and the decrease of the rate of cAMP generation [51,52,53]. Protein Gi activation induces the signaling through mitogen-activated protein kinase (MAPK) pathway that is well known to increase IFN-γ production [54,55,56,57]

β3-Ars, in contrast to β1-AR and β2-AR, couple with inhibitory G proteins and induce negative inotropic effects through the NO synthase/cyclic guanosine monophosphate/protein kinase G (NOS/cGMP/PKG) pathway. The precise role of β3-AR signaling within cardiomyocytes in promoting cardiac remodeling, which is not yet cleat in cells of the immune system. β3-ARs induce the expression of TGF-β1 in the culture of primary neonatal rat cardiomyocytes (NRCMs) by activating c-Jun, a subunit of AP-1, a critical transcription factor for TGFβ1. AP-1 specifically recognizes and binds to the TGFβ1 promoter, increasing TGFβ1 transcription [58]. Transforming growth factor β1 (TGFβ1) is a crucial cytokine mediating cardiac remodeling, and plays a causal role in the progression of heart failure.

Like β-AR, α-AR belongs to the GPCR superfamily. Activation of α1-AR fosters release of Ca++ as the second messenger from non-mitochondrial pools and protein kinase C (PKC) activation [59,60,61]. Some studies have investigated the expression of α1-AR by RT-PCR in mononuclear cells isolated from human primary and secondary lymphoid organs leading to different results.

Low expression of α1d-AR mRNA and a higher expression of α1a- and α1b-AR mRNA were found in human spleen primary lymphoid organs and thymus, but not in peripheral blood mononuclear cells (PBMC) [62,63,64,65].

This discrepancy could be related to the differential expression of these receptors during white blood cells maturation. It is likely that at early stages, leukocytes express α1-ARs, but not when mature cells enter the peripheral circulation.

Similarly, dendritic cells maturation is associated with changes in the expression of α1-AR. Immature dendritic cells express mRNA encoding the α1b-AR, lost in mature dendritic cells [66].

#### 2.2.2. Effects of AR Activation on the Innate Immune System

Adrenergic receptors’ expression and function have been well characterized in granulocytes, monocytes/macrophages, and dendritic cells, where β2-AR are usually the most expressed adrenoceptor and are considered as the primary receptor mediating the neuro-immunomodulation.

Adrenergic modulation of granulocytes has been examined mostly in neutrophils, but later evidence suggests that adrenergic agents play an essential role in the regulation of the activity of all granulocyte subtypes.

Monocytes and macrophages, together with DC, constitute the mononuclear phagocyte system, which plays a key role maintaining tissue integrity during development, its restoration after injury, as well as the initiation and resolution of innate and adaptive immunity. These cells respond rapidly to inflammatory signals by moving into the affected tissue and differentiating into macrophages and dendritic cells. Macrophages are classified in CD14++CD16−, M1 human macrophages or intermediates CD14++CD16+ cells, M2 human macrophages. Their role after myocardial infarction has been widely investigated. After ischemic injury there are two consecutive phases which M1 to M2 prevalence prompts transition: (1) inflammation, characterized by the prevalent activation of neutrophils and CD14++CD16− macrophages, and (2) restoration, characterized by fibrosis and neo-angiogenesis [67]. The level of β-ARs expression on human monocytes varies according to the different physiological and pathological scenarios [68,69]. The functional consequences of β-AR activation on human monocytes are usually anti-inflammatory and immunosuppressive and include: inhibition of oxygen radicals production [70], upregulation of TNF receptors, and inhibition of TNF [71]. Noradrenaline and adrenaline may have both pro- and anti-inflammatory effects on human monocytes. Recently the β2-AR antagonist propranolol has been shown to reduce circulating immunosuppressive M2b monocytes in severely burnt children, resulting in a reduction of their susceptibility to opportunistic infections [72]. Further studies are needed to understand if and how this receptor can be used to modulate macrophages activation and polarization toward an M1 and M2 phenotype improving cardiac remodeling.

Interaction of α2-AR and catecholamines has been demonstrated to be responsible for increasing TNF-α release [73,74,75]. TNF-α induces increased baseline catabolism via stimulating apoptosis, by activating the caspases pathway, which could contribute to the so-called "cardiac cachexia" [76]. TNF-α also induces myocyte necrosis via a cytotoxic mechanism related to the complement pathway, increasing NO synthase, and the local production of free radicals [77]. Elevated local TNF-α levels in the infarcted myocardium contribute to acute myocardial rupture and chronic left ventricle dysfunction by inducing an excessive local inflammatory response, matrix and collagen degradation, increasing matrix metalloproteinase activity, and apoptosis [78]. In animal models, cardiac-specific overexpression of TNF-α is responsible for the development of dilated cardiomyopathy [79], and systemic administration of TNF-α at some plasma concentrations comparable to those found in patients with congestive heart failure (CHF), have been shown to induce a dilated-cardiomyopathy-like phenotype in animal models [80].

Macrophages together with endothelial cells, leukocytes, and cardiomyocytes synthesize NFk-b, a transcription factor that regulates several proinflammatory substances and may be activated by multiple stimuli such as hypoxia, reactive O_2_ species, bacterial endotoxins, cytokines and others [77]. The myocardial tissue of patients with HF of different etiologies exhibits overexpression of this molecule and of the genes that it regulates, such as that associated with the synthesis of TNF-α, NO, leukocyte-adhesion molecules, and metalloproteinases [81].

Dendritic cells (DC) are specialized antigen-processing and presenting cells; when immature they have a robust phagocytic activity, developing a high cytokine production capacity when mature. DC circulate in the blood and migrate from tissues to lymphoid organs, regulating T cell responses [82]. Activation of β2-AR in DC increases intracellular cAMP and inhibits IL-12 production, resulting in the inhibition of Th1 and promotion of Th2 differentiation [83].

On the other hand, activation of β1-AR signaling in DC via the arrestin2-PI3K-MMP9/CCR7 pathway inhibits their migration (Figure 1) [84].

### 2.3. Natriuretic Peptides System and Immunomodulation

The system of natriuretic peptides is known to exert opposite effects on RAAS and SNS, and then potentially counteract their unfavorable consequences on the cardiovascular system during HF. So far, we recognize three natriuretic peptides: ANP, BNP, and CNP. All natriuretic peptides are synthesized as preprohormones.

Atrial myocytes secrete ANP in response to atrial wall stretching resulting from increased intravascular volume, endothelin, angiotensin, and arginine-vasopressin. Ventricular BNP production is regulated by cardiac wall stretching resulting from volume overload.

CNP is the most highly expressed natriuretic peptide in the brain and is found in high concentrations in chondrocytes and cytokine-exposed endothelial cells. CNP does not have a “natriuretic” action, and its plasma levels are not altered in heart disease such as congestive heart failure.

Natriuretic peptides elicit their physiological responses through the synthesis of cGMP, which acts as the second messenger. PKGs, serine, and threonine kinases are then activated by cGMP binding. The evidence that natriuretic peptides are expressed in immune organs and cells and that immunoregulatory agents regulate their expression and activity led to the hypothesis that they have a role in both innate and adaptive immune systems.

It has been reported that ANP is released within minutes of balloon occlusion of a major coronary artery for angioplasty so that it may contribute to the pathophysiological changes occurring in ischemia-reflow processes [85]. ANP is a potent signal for activating polymorphonucleated cells (PMN) respiration burst to secrete superoxide anion [86]. ANP also affects the phagocytic activity of macrophages; in particular, at physiological concentrations, ANP stimulates IgG uptake, promotes phagocytosis and killing of pathogens [87].

ANP can also interfere with the cytokines/chemokines system by reducing TNF-α secretion by macrophages [88] and interfering with the effects of TNF-α on the endothelium, with reduced expression of endothelial adhesion molecules, TNF-α-induced vascular permeability, and reduction in nitric oxide (NO) production through iNOS inhibition. ANP has also been reported to augment natural killer cell activity in vitro [89,90,91].

Dendritic cells (DC) have also been reported to express ANP receptors. In these cells, ANP regulates the balance between TH1 and TH2 responses and attenuating the exacerbated TH1 responses, leading to fibrosis and necrosis of the surrounding tissue [92].

The most studied among the natriuretic peptides is ANP, but many pieces of evidence about possible BNP’s immunomodulatory activity are emerging.

BNP may present both anti- and proinflammatory actions. It can stimulate ROS production through NADPH oxidase and increasing NO2 release in macrophages. It also can increase arachidonic acid release leading to an increase in LTB4 and PGE2. PGE2 can act both as a proinflammatory and anti-inflammatory mediator depending on the context. It can also modulate cytokine production in several cell types, up-regulating the production of IL-10 by macrophages and inhibiting IL-12 and TNF-α release by dendritic cells (DCs), suggesting an anti-inflammatory cytokines profile induction. BNP can also increase the migratory activity of macrophages after 24 h of stimulation [93]

CNP shows cytoprotective effects in experiments demonstrating the reduction in myocardial ischemia/reperfusion injury [94]. CNP infusion significantly attenuates necrosis and infiltration of CD68-positive inflammatory cells in myocarditis.

CNP suppresses the production of pro-inflammatory COX 2-derived PGE2, which possesses anti-inflammatory actions in the vasculature, similar to endothelium-derived NO, and decreases platelets aggregation and leucocytes rolling/adhesion through its action on P-selectin [95].

In summary, natriuretic peptides are supposed to be essential regulators of inflammatory mediators that may have broad implications in heart failure (Figure 1).

## 3. Mechanism of Immune Modulation in Heart Failure with Preserved Ejection Fraction

HFpEF is associated with numerous comorbidities and risk factors like old age, female sex, hypertension, diabetes mellitus, obesity, and atrial fibrillation, however, its underlying pathophysiology remains subject to debate. 

HFpEF often presents with diastolic dysfunction including delayed early relaxation, myocardial and myocyte stiffening, and changes in filling dynamics, but other mechanisms playing an important role are the limited heart rate augmentation (chronotropic incompetence), reduced peripheral vascular dilation, altered myocardial energetics, renal insufficiency, the accumulation of extracellular matrix, and EndMT [6,96]. The multitude of HFpEF comorbidities may contribute to a proinflammatory state; circulating inflammatory cytokines such as interleukin 6, tumor necrosis factor α, soluble ST2, and pentraxin 3 are elevated in HFpEF, indeed [97]. Systemic inflammation could lead to endothelial dysfunction supported by higher expression of vascular cell adhesion molecules such as VCAM-1, E-selectin, and ROS. TGFβ signaling may also be increased in HFpEF myocardiocytes [97].

The pro-inflammatory state in HFpEF is responsible for the increased oxidative stress through the activation and migration of leucocytes, and increased superoxide production inducing hypertrophy. The interaction between NO and superoxide generates the peroxynitrite reducing the NO bioavailability levels resulting in low intracellular cGMP and reduced PKG activity. A deficient NO-sGC-cGMP pathway increases diastolic cytosolic Ca^2+^ and delays myocardial relaxation. Myocardial stiffness also increases through a deficient NO-sGC-cGMP pathway [97].

The potential mechanisms for the altered matrix structure accumulation include inflammation and neurohumoral stimuli such as the RAAS [5].

Similarly to HFrEF, patients with HFpEF present increased plasma renin activity but not norepinephrine or ANP levels. RAAS activation is, indeed, considered more relevant than other neurohormonal mechanisms in HFpEF-related fibrosis [98].

Some anatomopathological studies on endomyocardial biopsies and analyses of inflammatory cell markers suggest that increased oxidative stress and depressed NO signaling, resulting in inflammation, play a crucial role in HFpEF. These pieces of evidence are also supported by angiotensin-converting enzyme (ACE) inhibitors, which can revert this process by increasing nitric oxide release [97]. Aldosterone and its mineralocorticoid receptors are also potentially involved in this context. Administration of spironolactone to patients with HFpEF has been shown to relieve symptoms and decrease left atrial area, with a beneficial effect on diastolic function [99,100,101]. However, the effects of mineralocorticoid on regulating the immune system in HFpEF has not been demonstrated yet. Nonetheless, the cross-talk between the neurohumoral pathways and the immune system have not been studied adequately in heart failure with preserved systolic function. The evidence is currently available for the cross-talk between the renin-angiotensin and immune system, but none for the adrenergic system and natriuretic peptide system. However, these and other neurohumoral pathways may play a fundamental role in the pathogenesis of the HFpEF, with still unexplored mechanisms of inflammatory process regulation.

## 4. Therapeutic Translation

Beta-blocker therapy remains one of the fascinating issues in heart failure treatment. Clinical trials assessing the use of beta-blockers (BBs) in CHF have established benefits in both mortality and morbidity outcomes [102,103]. However, it appears that it is not a class effect, but significant differences exist between individual BBs—considering that not all BBs have shown similar benefits on clinical endpoints. These agents are now the cornerstone of current HF treatments. 

The effects on the immune system of these drugs confirm the role of the adrenergic system in immunomodulation.

In mice with viral myocarditis, carvedilol has been shown to modulate the production of IL-12 and IFN-γ, improving their survival [104]. It has also been shown in humans that it decreases the production of ROS, such as H_2_O_2_, which is responsible for driving calcium overload in HF. Moreover, this drug suppresses plasma levels of TNF-α and IL-6 in both ischemic and non-ischemic dilated cardiomyopathy patients [105]. 

Metoprolol tartrate treatment is responsible for the increase in T cells, natural killer cells, and interleukin-2 receptor density. These changes correlate to the increases in ejection fraction [106].

Similarly, propranolol treatment increases the number of circulating T cells with enhanced lymphocyte proliferation and IL-2 formation. Despite it not altering the number of circulating NK cells, their activity is reduced after propranolol treatment.

Since 1990, Angiotensin-converting enzyme inhibitors (ACE-I) and AT1 receptor blockers have become the routine practice in MI management [107,108].

More recent studies have revealed the beneficial effects of RAS inhibition in mediating post-MI inflammation, which is responsible for adverse remodeling and heart failure. Animal studies showed that both losartan and perindopril decrease inflammatory gene expression and increase the expression of the anti-inflammatory cytokine IL-10 [109,110]. Losartan and irbesartan have been shown to attenuate cardiac fibrosis through the inhibition of EndM transition via TGF-β/SMAD [16]. Reduction of TGF-β levels has also been associated with ACE inhibitors administration [40,111,112].

The sacubitril/valsartan association has recently been demonstrated to be superior compared to ACE-I (Enalapril) in reducing cardiovascular death or hospitalizations for HF. Sacubitril is responsible for the inhibition of neprilysin which degrades natriuretic peptides, thus allowing natriuretic peptides to persist longer and promote vasodilation, diuresis, natriuresis, and prevent cardiac hypertrophy. Sacubitril is used in combination with an angiotensin receptor blocker (ARB) that acts on the renin-angiotensin-aldosterone system, prevents vasoconstriction, and decreases both aldosterone secretion and renal reabsorption of sodium.

By simultaneously inhibiting the angiotensin receptor and neprilysin, the sacubitril/valsartan association improves cardiac dysfunction, hypertension, cardiovascular injury, and ischemic brain damage in experimental and clinical studies [113]. In a recent animal model, the association of sacubitril and valsartan has been demonstrated to have an important role in modulating the immune system. Indeed, this association significantly decreases mRNA expression of IL-1 β and IL-6 in the infarcted region three days after MI, and the IL-1β and IL-6 protein production by macrophages (Figure 2). These pro-inflammatory cytokines signaling pathways can regulate MMP-9 expression of macrophages, involved in tissue repair, remodeling, and development of cardiac rupture after myocardial infarction [114].

IL-1β and TNF-α have been shown to depress myocardial contractility. This effect may be due to uncoupling of β-adrenergic signaling, increase in cardiac nitric oxide, or alterations in intracellular calcium homeostasis [115].

Proinflammatory cytokines (interleukin-1, -2, -6, and tumor necrosis factor) and chemokines are not only markers of immune activation but may also play a pathogenic role in CHF, and are involved in cardiac depression and the progression of heart failure.

The effort to define the immune system’s role has opened new perspectives of therapeutic strategies, such as anti-cytokine drugs, to treat CHF.

Trials investigating the effects of etanercept as a treatment of heart failure failed to demonstrate beneficial effects [116,117]. 

Therapy with intravenous immunoglobulins (IVIG) has been demonstrated to significantly improve left ventricle ejection fraction (LVEF) by 5% in CHF patients, independent of the etiology of heart failure. They also improve some hemodynamic variables (pulmonary capillary wedge pressure) and exercise capacity, reducing plasma levels of amino-terminal pro-atrial natriuretic peptide [82]. Unfortunately, the effects decrease one year after terminating therapy [118]. 

The most promising immunomodulator drug in CHF treatment is the pentoxifylline, which inhibits the production of tumor necrosis factor-α resulting in a reduction of its negative inotropic properties, and reduces the expression of Fas/APO-1—an apoptosis-signaling surface receptor known to trigger programmed cell death in a variety of cell types [119,120].

In patients with heart failure already receiving treatment with ACE inhibitors and beta-blockers, the addition of pentoxifylline is associated with a significant improvement in symptoms and left ventricular function [121,122]. 

Immunomodulatory drugs, in addition to conventional HF treatments, could in the future, lead to an improvement in clinical symptoms and survival, of patients with heart failure.

## 5. Conclusions

All neurohumoral pathways involved in HF are responsible for “cardiac remodeling” referred to the changes in cardiac mass, volume, and shape, and in heart composition that occurs after a cardiac injury or abnormal hemodynamic loading conditions. These changes concern myocyte biology, energetics, metabolism, progressive loss of myocytes through apoptosis, autophagic cell death, and necrosis are important mechanisms and the neurohormonal regulation of immune-system plays a pivotal role in this complex cascade.

Interventions aimed at immune modulation might be useful in treating heart failure, even if a large body of research on the activation of immune pathways involves primarily animal studies. So far, first line drugs used in HF remain beta blockers and drugs targeting the RAAS system, which appear to accomplish positive effects on cardiac remodeling by modulating the immune system. The recent introduction of natriuretic peptides system (ARNI) modulators in HF therapies, such as sacubitril, may have also beneficial effects on cardiac remodeling by modulating inflammatory responses and potentially adding new knowledge on the neurohormonal-immune system cross-talk. Nonetheless, strategies designed as immunomodulatory therapy are a promising approach for the treatment of heart failure.

## Figures and Tables

**Figure 1 ijms-20-01698-f001:**
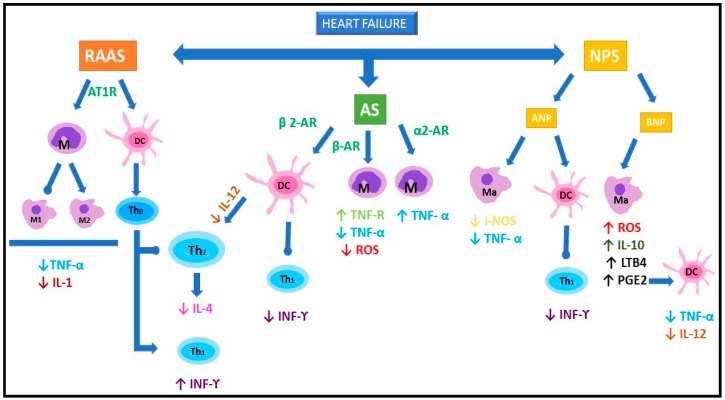
Effects of the neurohormonal system on the immune system. RAAS system: Angiotensin II (Ang II) regulates macrophage polarization, promoting M2 macrophages phenotype (M2). The consequent inhibition of M1 phenotype leads to a reduced release of TNF-α and IL-1. Moreover, Ang II preferentially promotes Th1 lymphocytes (Th1) differentiation through regulation of dendritic cells (DC), leading to increased INF-ϒ levels. Inhibition of Th2 reduces IL-4 production and release. Adrenergic System (AS)*:* β and α2-AR exert opposite effects on TNF-α release. β-AR activation in macrophages is responsible for the inhibition of ROS production, increase in TNF-α receptors, and reduction in TNF-α release. However, α2-AR increase TNF-α release. In dendritic cells, activation of β2-AR inhibits IL-12 production, resulting in inhibition of Th1 and promotion of Th2 differentiation. Natriuretic Peptide system (NPS): ANP attenuates inflammatory responses by the reduction of TNF-α release and nitric oxide production by iNOS. It also reduces the exacerbated TH1 responses. BNP in macrophages can stimulate ROS production. Its action is also the up-regulation of IL-10, LTB4, and PGE2. PGE2 is responsible of the inhibition of IL-12 and TNF-α release by dendritic cells. Abbreviations: ↑ up-regulation;↓ down-regulation; 
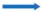
 stimulation; 
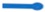
 inhibition; RAAS: renin-angiotensin-aldosterone system; AS: adrenergic system; NPS: natriuretic peptides system; ANP: atrial natriuretic peptide; BNP: brain natriuretic peptide; AT1R: angiotensin receptor 1; AR: adrenergic receptor; M: monocyte; Ma: macrophage; M1: type 1 macrophage; M2: type 2 macrophage; DC: dendritic cell; Th0: non polarized lymphocyte; Th1: Th1 lymphocytes; Th2: Th2 lymphocytes; TNF: tumor necrosis factor; IL: interleukin; i-NOS: inducible nitric oxide synthase; ROS: radical oxygen species; LTB4: leukotriene B4; PGE2:prostaglandin E2; IFN: interferon.

**Figure 2 ijms-20-01698-f002:**
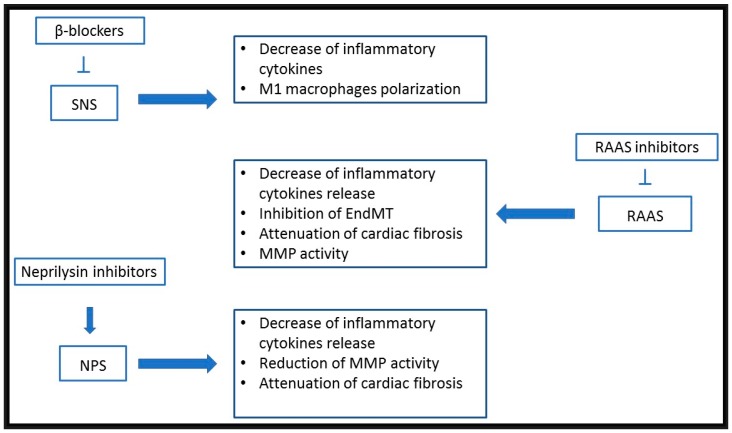
Immunomodulatory effects of commonly used heart failure drugs. Beta blockers reduce inflammatory cytokines release and M1 macrophages polarization. Each beta-blocker molecule can have others specific action in immunoregulation. RAAS inhibitors decrease the inflammatory cytokines release reducing matrix metalloproteinases (MMP) activity and inhibit the EndMT attenuating cardiac fibrosis. Sacubitril reduces inflammatory cytokines release and MMP activity attenuating cardiac fibrosis. Abbreviations: RAAS: renin-angiotensin-aldosterone system; MMP: matrix metalloproteinases; EndMT: endothelial to mesenchymal transition.

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
