# Peer review of "Cross-Talk between Neurohormonal Pathways and the Immune System in Heart Failure: A Review of the Literature"

_ijms, 2019, doi:10.3390/ijms20071698_

Round 1
Reviewer 1 Report
General comments
This review is generally on the communication between the neurohumoral axis and immune system activation in heart failure syndromes. Pathophysiology of heart failure is, indeed, very complex and involves activation of numerous pathways such as adrenergic system, neurohumoral systems, RAAS, arginine-vasopressin, and so on. Furthermore, elucidation of proinflammatory pathways and immunologic involvement in heart failure has been of great research interest lately, therefore, this review is well-timed and covers currently a pertinent topic in the field. However, certain things would need to be taken into account and amended respectively before this paper is considered for possible publication. Specific comments are outlined below.
Ethical approval
Not applicable.
Statistical methods
Not applicable.
Specific comments:
1. Abstract – in „polymorphonucleated cells“ I suppose. Please correct.
2. In Introduction, when defining heart failure please insert Braunwald. JACC Heart Fail. 2013 reference – review article on HF.
3. A review of this kind would greatly benefit from relevant figures that could enhance this manuscript greatly and would increase readership interest and a possibility of further citations. This review has potential, but in order for it to be maximized, I would strongly encourage authors to create relevant figures that would depict:
a. Interaction of neurohumoral pathways on immune cell lineages... e.g. activation of angiotensin II with macrophages à M2 macrophage phenotype promotion, etc. Please use the same rationale to depict other relevant pathways discussed in this paper
b. Similarly, figures could be provided to display relevant pathways regarding RAAS and immune system, nervous sympathetic system and immune system and, finally, natriuretic peptides system and immune system
4. Line 129 – please change „dis-adaptive“ to „maladaptive“
5. Line 152 – please change „differently by B1-AR and B2-AR“ to „in contrast to B1-AR and B2-AR“ and replace „induce adverse“ with „induce negative“ inotropic effects...
6. One summarizing figure showing immunomodulatory effects of beta blockers (carvedilol, metoprolol, propranolol), ACE/ARBS and ARNIs would be very welcome
7. Line 327-330 – Authors state that most promising immunomodulator drug in CHF treatment is pentoxifylline, yet, they do not provide information on which modulatory effects in terms of interaction with the immune system are exerted by the pentoxifylline
8. While the scope of the work made by the authors is relevant, one thing that is largely missing from this review is an appreciation of different HF phenotypes that are being studied in modern cardiovascular science. The pathophysiology, etiology, and prognosis of HF with reduced ejection fraction differ in many ways from those that in preserved systolic function (also known as HFpEF). Authors should create and dedicate a relevant paragraph in which they would highlight relevant differences in terms of immunomodulation and inflammation between these two HF phenotypes. This is extremely relevant, rather than perceiving HF as a purely dilated cardiomyopathy which we all know is too simplistic and not true. In fact, HFpEF might account to up to 50% of all newly-diagnosed HF cases, according to latest registry data, especially those gathered in Europe
9. To make this review more relevant, authors should tackle the issue of endothelial-mesenchymal transition (EndoMT pathway) as a relevant pathway in HF that leads to fibrosis and LV dysfunction... these are novel pathophysiological targets that are relevant in terms of preserved HF phenotype development
10. Finally, to make this review interesting to clinicians as well, authors should provide some background on comorbidities and effects of comorbidities on HF pathophysiology, at least in a small paragraph and how they can enhance adverse immunologic responses (atrial fibrillation, obesity, exercise intolerance, arterial hypertension, etc.)
11. Sentence 331-332 should be rewritten because it is not clear in the current form
Author Response
1. Abstract – in „polymorphonucleated cells“ I suppose. Please correct.
We corrected the previous wrong version.
2. In Introduction, when defining heart failure please insert Braunwald. JACC Heart Fail. 2013 reference – review article on HF.
Suggested reference has been added.
3. A review of this kind would greatly benefit from relevant figures that could enhance this manuscript greatly and would increase readership interest and a possibility of further citations. This review has potential, but in order for it to be maximized, I would strongly encourage authors to create relevant figures that would depict:
a. Interaction of neurohumoral pathways on immune cell lineages... e.g. activation of angiotensin II with macrophages à M2 macrophage phenotype promotion, etc. Please use the same rationale to depict other relevant pathways discussed in this paper
b. Similarly, figures could be provided to display relevant pathways regarding RAAS and immune system, nervous sympathetic system and immune system and, finally, natriuretic peptides system and immune system
As suggested, a figure that summerizes relevant pathways regarding RAAS and immune system, nervous sympathetic system and immune system and, finally, natriuretic peptides system and immune system has been added. (Figure 1)
4. Line 129 – please change „dis-adaptive“ to „maladaptive“
“Maladaptive” now replaces “dis-adaptive”
5. Line 152 – please change „differently by B1-AR and B2-AR“ to „in contrast to B1-AR and B2-AR“ and replace „induce adverse“ with „induce negative“ inotropic effects...
All suggested corrections have been made.
6. One summarizing figure showing immunomodulatory effects of beta blockers (carvedilol, metoprolol, propranolol), ACE/ARBS and ARNIs would be very welcome
As suggested, a figure that summerizes immunomodulatory effects of drugs has been added (Figure 2).
7. Line 327-330 – Authors state that most promising immunomodulator drug in CHF treatment is pentoxifylline, yet, they do not provide information on which modulatory effects in terms of interaction with the immune system are exerted by the pentoxifylline
More informations regarding the immunomodulatory effects of pentoxifylline have been added.
8. While the scope of the work made by the authors is relevant, one thing that is largely missing from this review is an appreciation of different HF phenotypes that are being studied in modern cardiovascular science. The pathophysiology, etiology, and prognosis of HF with reduced ejection fraction differ in many ways from those that in preserved systolic function (also known as HFpEF). Authors should create and dedicate a relevant paragraph in which they would highlight relevant differences in terms of immunomodulation and inflammation between these two HF phenotypes. This is extremely relevant, rather than perceiving HF as a purely dilated cardiomyopathy which we all know is too simplistic and not true. In fact, HFpEF might account to up to 50% of all newly-diagnosed HF cases, according to latest registry data, especially those gathered in Europe
As suggested, Introduction has been enterily revised and the differences between HFrEF and HFpEF have been highlighted.
9. To make this review more relevant, authors should tackle the issue of endothelial-mesenchymal transition (EndoMT pathway) as a relevant pathway in HF that leads to fibrosis and LV dysfunction... these are novel pathophysiological targets that are relevant in terms of preserved HF phenotype development.
As suggested, in Introduction section, the role of endothelial-mesenchymal transition (EndoMT pathway) as a relevant pathway in HF has been highlighted.
10. Finally, to make this review interesting to clinicians as well, authors should provide some background on comorbidities and effects of comorbidities on HF pathophysiology, at least in a small paragraph and how they can enhance adverse immunologic responses (atrial fibrillation, obesity, exercise intolerance, arterial hypertension, etc.)
As suggested, we provide some background on comorbidities on HF, so a new paragraph, entitled “3.Mechanism of immune modulation in Heart failure with preserved Ejection Fraction” has been added
11. Sentence 331-332 should be rewritten because it is not clear in the current form.
The sentence has been rewritted.

Reviewer 2 Report
The present manuscript by De Angelis et al. is an interesting and timely literature review article of the currently "hot" research topic of the interplay between hormones and the immune system in heart failure. I have a few comments to help the authors improve the quality of their manuscript:
1) The review covers cellular (in vitro) and animal studies quite thoroughly but lacks clinical perspectives. The authors should expand their discussion of relevant clinical studies, i.e. elaborate more on whatever findings are available in actual heart failure patients, as well as on the main clinical implications of the studies they have reviewed.
2) I understand that inclusion of some old papers is necessary to give a comprehensive account of the literature and to also add a historic perspective but the vast majority of the citations are way too old (>15 years!). Please replace as many of those references as possible with more recent ones.
3) In the same vein, when discussing the roles of adrenergic receptors and of aldosterone in immunomodulation of heart failure, a few important relevant reviews are missing and should be cited: Drug Des Devel Ther. 2013;7:1209-22; Curr Heart Fail Rep. 2015;12:130-40; Circ Res. 2013;113:739-53; Pharmacol Res. 2017;125:14-20.
4) Line 131: this sentence is false: there are actually nine mammalian AR subtypes in total, three alpha1, three alpha2, and three beta (e.g. see: Eur J Pharmacol. 2015;763:143-8).
5) Line 221: given that beta-arrestin2 is arrestin-3 (and beta-arrestin1 is arrestin-2), this sentence should read "arrestin3-PI3K-MMP9/CCR7" instead of "arrestin2-PI3K-MMP9/CCR7", which it reads now. Also, although arrestins are mentioned at various points throughout the text, the authors do not provide a (brief) introduction of these proteins, so that the non-familiar reader knows what their biological functions are. To that end, the following two reviews covering specifically the cardiovascular arrestins are recommended: Prog Mol Biol Transl Sci. 2018;159:27-57 & Int Rev Cell Mol Biol. 2018;339:41-61.
6) Inclusion of a schematic illustration summarizing the most important signaling pathways discussed in the manuscript would make the review more appealing aesthetically but also more readily understandable to the reader.
Author Response
1) The review covers cellular (in vitro) and animal studies quite thoroughly but lacks clinical perspectives. The authors should expand their discussion of relevant clinical studies, i.e. elaborate more on whatever findings are available in actual heart failure patients, as well as on the main clinical implications of the studies they have reviewed.
As suggested, cues trials have been added, and the paragraph entitled “Therapeutic translation” has been expanded.
2) I understand that inclusion of some old papers is necessary to give a comprehensive account of the literature and to also add a historic perspective but the vast majority of the citations are way too old (>15 years!). Please replace as many of those references as possible with more recent ones.
Where possible, more recent references have been added.
3) In the same vein, when discussing the roles of adrenergic receptors and of aldosterone in immunomodulation of heart failure, a few important relevant reviews are missing and should be cited: Drug Des Devel Ther. 2013;7:1209-22; Curr Heart Fail Rep. 2015;12:130-40; Circ Res. 2013;113:739-53; Pharmacol Res. 2017;125:14-20.
Suggested references have been added in the text.
4) Line 131: this sentence is false: there are actually nine mammalian AR subtypes in total, three alpha1, three alpha2, and three beta (e.g. see: Eur J Pharmacol. 2015;763:143-8).
As suggested, the reference has been corrected.
5) Line 221: given that beta-arrestin2 is arrestin-3 (and beta-arrestin1 is arrestin-2), this sentence should read "arrestin3-PI3K-MMP9/CCR7" instead of "arrestin2-PI3K-MMP9/CCR7", which it reads now. Also, although arrestins are mentioned at various points throughout the text, the authors do not provide a (brief) introduction of these proteins, so that the non-familiar reader knows what their biological functions are. To that end, the following two reviews covering specifically the cardiovascular arrestins are recommended: Prog Mol Biol Transl Sci. 2018;159:27-57 & Int Rev Cell Mol Biol. 2018;339:41-61.
As suggested, a brief introduction for arrestins has been added and the suggested reference has been added.
6) Inclusion of a schematic illustration summarizing the most important signaling pathways discussed in the manuscript would make the review more appealing aesthetically but also more readily understandable to the reader.
As suggested, two figures have been added. Figure 1 summerizes relevant pathways regarding RAAS and immune system, nervous sympathetic system and immune system and, finally, natriuretic peptides system and immune system; figure 2 summerizes immunomodulatory effects of drugs.
Round 2
Reviewer 1 Report
Authors have addressed all my comments and concerns effectively.
I would wish to congratulate them on well-made figures and expansion of the text in relevant aspects, therefore, the manuscript has been improved substantially.
Reviewer 2 Report
The manuscript has been substantially improved. No further comments.